# Fabrication and Mechanism of Pickering Emulsions Stability over a Broad pH Range Using Tartary Buckwheat Protein–Sodium Alginate Composite Particles

**DOI:** 10.3390/foods14193429

**Published:** 2025-10-05

**Authors:** Yu Song, Xueli Shen, Gangyue Zhou, Xia Xu, Yanan Cao, Wei Li, Yichen Hu, Jianglin Zhao, Dingtao Wu, Zunxi Huang, Liang Zou

**Affiliations:** 1Key Laboratory of Coarse Cereal Processing (Ministry of Agriculture and Rural Affairs), Sichuan Engineering & Technology Research Center of Coarse Cereal Industrialization, School of Food and Biological Engineering, Chengdu University, Chengdu 610106, China; songyu@cdu.edu.cn (Y.S.); shenxueli2025@126.com (X.S.); zhougangyue2025@126.com (G.Z.); caoyanan@cdu.edu.cn (Y.C.); huyichen0323@126.com (Y.H.); zhaojianglin@cdu.edu.cn (J.Z.); 2Key Laboratory of Yunnan for Biomass Energy and Biotechnology of Environment, School of Life Science, Yunnan Normal University, Kunming 650500, China; huangzunxi@163.com; 3School of Leisure Tourism, Chengdu Agricultural College, Chengdu 611130, China; xuxia202509@126.com; 4School of Basic Medical Sciences, Chengdu University, Chengdu 610106, China; liwei@cdu.edu.cn

**Keywords:** tartary buckwheat protein, sodium alginate, complex, pickering emulsion, stability, pH stability

## Abstract

In this study, the insufficient ability of tartary buckwheat protein (TBP) to stabilize Pickering emulsions was addressed by preparing TBP–sodium alginate (SA) composite particles via cross-linking and systematic optimization of the preparation parameters. The results showed that at a pH of 9.0 with 1.0% (*w*/*v*) TBP and 0.2% (*w*/*v*) SA, the zeta potential of the prepared TBP–SA composite particles was significantly more negative, and the particle size was significantly larger, than those of TBP, while emulsifying activity index and emulsifying stability index increased to 53.76 m^2^/g and 78.78%, respectively. Scanning electron microscopy confirmed the formation of a dense network structure; differential scanning calorimetry revealed a thermal denaturation temperature of 83 °C. Fourier transform infrared spectroscopy and surface hydrophobicity results indicated that the complex was formed primarily through hydrogen bonding and hydrophobic interactions between TBP and SA, which induced conformational changes in the protein. The Pickering emulsion prepared with 5% (*w*/*v*) TBP–SA composite particles and 60% (φ) oil phase was stable during 4-month storage, at a high temperature of 75 °C, high salt conditions of 600 mM, and pH of 3.0–9.0. The stabilization mechanisms may involve: (1) strong electrostatic repulsion provided by the highly negative zeta potential; (2) steric hindrance and mechanical strength imparted by the dense interfacial network; and (3) restriction of droplet mobility due to SA-induced gelation.

## 1. Introduction

Pickering emulsions have a wide range of applications in various fields, including food and pharmaceutical products, owing to their unique stabilization mechanism involving solid particles [1,2,3]. The stabilizers, including proteins [4], polysaccharides [5], and fat crystals [6], used in Pickering emulsions have evolved from synthetic inorganic particles to edible organic particles. Tartary buckwheat protein (TBP) [7] has garnered significant attention due to its various bioactivities, including antioxidant activity [8], intestinal flora modulation [9], cholesterol reduction [10], and lipid metabolism regulation [11]. Although TBP is effective in terms of water-holding capacity, emulsifying properties, and foaming ability [12], studies have shown that Pickering emulsions prepared with TBP are unstable, with a strong tendency to cream. Thus, the ability of TBP to stabilize Pickering emulsions needs to be improved. This instability is due to the low surface hydrophobicity of TBP and its easy aggregation near its isoelectric point (pI ≈ 4.0–5.0) [13,14,15], which hinders its effective adsorption at the oil–water interface and the formation of a stable interfacial film.

Polysaccharides can form complexes with proteins through hydrogen bonding, electrostatic interactions, and other mechanisms [16,17,18], representing an effective strategy to improve the emulsifying properties of proteins and enhance emulsion stability [19,20]. Sodium alginate (SA), an anionic polysaccharide, exhibits excellent water absorption and thickening capabilities. It can be synergistically assembled with proteins through non-covalent interactions, thus enhancing the interfacial stability of the particles [21]. Previous studies have reported that complexes formed with whey protein [22], quinoa protein [23], egg white protein [24], etc., can significantly improve the stability and functional properties of emulsions. However, to date, no study has been reported on the preparation of TBP–SA particles and their use in stabilizing Pickering emulsions, particularly in exploring their interfacial stabilization mechanism under different environmental conditions.

Therefore, this study hypothesizes that under alkaline conditions (pH 9.0), TBP and SA can form stable composite particles through electrostatic interactions and hydrogen bonding, thereby constructing Pickering emulsions with broad pH stability. In this study, TBP–SA nanocomposites were prepared by cross-linking TBP with SA, and their particle size, zeta potential, emulsification properties, and Fourier transform infrared (FTIR) spectral characteristics were examined. Furthermore, Pickering emulsions stabilized by TBP–SA composites were prepared. The effect of TBP–SA complexes on emulsion stability was assessed by analyzing emulsion droplet diameter, confocal laser scanning microscopy (CLSM) microstructure, creaming index, and other parameters. This work broadens the application of TBP in Pickering emulsions and provides insights for developing functional delivery systems.

## 2. Materials and Methods

### 2.1. Materials

Laboratory-prepared tartary buckwheat protein (TBP, 86.80 ± 0.62% as determined by the BCA method using a kit from Nanjing Jiancheng Bioengineering Institute, Nanjing, China), sodium alginate (SA, food grade) and Medium-chain triglycerides (MCT, food grade) were purchased from Shanghai Aladdin Biochemical Technology Co., Ltd. (Shanghai, China). Potassium bromide (spectrum pure) was supplied by Shanghai Aladdin Biochemical Technology Co., Ltd. (Shanghai, China). Sodium dodecyl sulfate (analytical grade) was sourced from Saiguo Biotechnology Co., Ltd. (Guangzhou, China). Sodium hydroxide (analytical grade) and hydrochloric acid (analytical grade) were purchased from Chengdu Kelong Chemical Co., Ltd. (Chengdu, China).

### 2.2. Preparation of TBP–SA Complexes

#### 2.2.1. Effect of Different pH on TBP–SA Complex Preparation

Stock solutions of 2% (*w*/*v*) TBP (dissolved in pH 7.0 PBS) and 0.4% (*w*/*v*) SA (dissolved in deionized water) were separately prepared and hydrated at 4 °C for 12 h. After ultrasonication, equal volumes (10 mL:10 mL) were mixed. The pH of the mixture was adjusted to 3, 5, 7, 9, and 11 using HCl (1 mol/L) or NaOH (1 mol/L), magnetically stirred for 30 min, and subjected to a second ultrasonication. Particle size and zeta potential were determined to investigate the effects of different pH values on TBP–SA complex preparation. Both ultrasonication steps were performed using a probe-type ultrasonicator (probe diameter 6 mm) in an ice-water bath with the temperature controlled (alarm temperature 60 °C). The specific parameters were: amplitude 76% (rated power 450 W), pulse mode (on 1.0 s, off 1.5 s), and a single treatment duration of 2 min.

#### 2.2.2. Effect of SA Concentration on TBP–SA Complex Preparation

Stock solutions with gradient concentrations of SA (0.06–0.8% (*w*/*v*)) were prepared and mixed with equal volumes of 2% (*w*/*v*) TBP (pH 9). After stirring for 30 min followed by ultrasonication, the effects of SA concentrations (0–0.4% (*w*/*v*)) on TBP–SA complex preparation were investigated using particle size, zeta potential, and emulsifying properties as evaluation indicators.

### 2.3. Determination of Particle Size and Zeta Potential of TBP–SA Complexes

Particle size and zeta potential were measured using a Nano-ZS nanoparticle analyzer (Malvern Instruments Ltd., Malvern, UK) following the method of Liu et al. [25,26]. TBP and TBP–SA complex samples were diluted with deionized water at a 1:50 (*v*/*v*) ratio and analyzed under constant temperature conditions at 25 ± 1 °C. For particle size measurement, the refractive index was set to 1.414; after a 120 s equilibration period, three parallel measurements were performed and averaged. For zeta potential determination, samples were equilibrated for 60 s followed by three parallel measurements. All tests were completed in automatic mode.

### 2.4. Determination of Emulsifying Properties of TBP–SA Complex

Emulsifying properties of the TBP–SA complex were determined following the previously reported method [27]. First, 3 mL of sample (TBP or TBP–SA) was mixed with 2 mL MCT oil, vortexed for 1 min, and then sheared at 15,000 rpm for 2 min to prepare the emulsion. Then, 100 μL of the emulsion was mixed with 9.9 mL of 0.1% SDS solution. The initial absorbance (A_0_) was measured at 500 nm, and A_10_ was measured after 10 min. The blank control was 100 μL of water + 9.9 mL of SDS solution. Emulsifying activity index (EAI) and emulsion stability index (ESI) were calculated using the following formulas:$$\mathrm{EAI}\;(m^{2}/g)\;=\;\frac{2\;\times \;2.303\;\times \;A_{0}\;}{10000\;\times \;(1\;-\;\Phi )\;\times \;C\;}\;\times \;\mathrm{DF}$$$$\mathrm{ESI}\left(\%\right)=\frac{A_{10}}{A_{0}}\;\times \;100$$
where C is the initial protein concentration (g/mL), i.e., 0.01 g/mL; Φ is the volume fraction of the oil phase (0.4), and DF is the dilution factor (1000).

### 2.5. Determination of Surface Hydrophobicity of TBP–SA Complexes

The surface hydrophobicity of TBP and TBP–SA was determined by bromophenol blue method [28]. 1 mL sample solution was fully mixed with 1 mL PBS (0.01 M, pH 7.0) and 200 μL BPB solution (1 mg/mL), and vortexed for 10 min. Subsequently, the mixture was centrifuged at 4000× *g* at 4 °C for 15 min. The supernatant was diluted 10 times with PBS, and PBS solution was used as a blank control. The absorbance was measured at 595 nm using an ultraviolet-visible spectrophotometer. The binding amount of bromophenol blue was calculated according to the following formula to characterize the relative surface hydrophobicity of the sample.$$BPB\;Combination\left(\mu g\right)=\frac{200\times (A_{0}-A_{1})}{A_{0}}$$

A_0_ is the absorbance of the blank control containing only PBS and BPB, and A_1_ is the absorbance of the sample to be tested.

### 2.6. Determination of Differential Scanning Calorimetry (DSC) for TBP–SA Complex

First, 4–5 mg of freeze-dried samples (TBP, SA, and TBP–SA complex) was placed into an aluminum crucible, tamped, and hermetically sealed. Thermograms were recorded using an empty crucible as a blank control under a nitrogen flow rate of 20 mL/min, scanning temperature range of 20–200 °C, and heating rate of 10 °C/min.

### 2.7. Determination of Fourier Transform Infrared Spectroscopy (FTIR) for TBP–SA Complex

Following the previously reported method [29], freeze-dried samples (TBP, SA, and TBP–SA complex) were thoroughly ground and mixed with potassium bromide at a 1:100 mass ratio. Then, they were pressed into pellets. A Fourier transform infrared spectrometer (PerkinElmer, Boston, MA, USA) was used to obtain the infrared spectral characteristics of the samples in the wavenumber range of 400–4000 cm^−1^ with a cumulative total of 64 scans. The collected spectra were subjected to baseline correction using the instrument’s software (OMNIC 9.2) before further analysis.

### 2.8. Scanning Electron Microscopy (SEM) Observation of TBP–SA

The prepared powder samples (TBP, SA, and TBP–SA) were placed on sample stages and sputter-coated with gold under vacuum to enhance conductivity. Subsequently, the samples were placed on the stage of a scanning electron microscope (Thermo Fisher Scientific, Waltham, MA, USA). By adjusting the field of view and magnification, the surface micromorphology of the samples was observed and analyzed, with structural features recorded using an image acquisition system. The specific parameters were as follows: accelerating voltage of 10 kV, electron beam spot size of 2.0, working distance of 10 mm, magnification of 10,000×, and vacuum degree of 6.67 × 10^−3^ Pa. Images were acquired using the secondary electron (SE) mode with an ETD detector.

### 2.9. Preparation of TBP–SA Complex-Stabilized Pickering Emulsion

The freeze-dried TBP–SA complex powder was dissolved in PBS (pH 7.0, 0.01 mol/L) to prepare aqueous complex solutions at concentrations of 0.5, 1, 3, and 5% (*w*/*v*). After the mixture was magnetically stirred for 30 min, its pH was adjusted to 9.0. MCT was added as the oil phase to the complex solution with oil phase fractions of 20, 40, 60, and 80%. Emulsification was performed using a High-speed shearing dispersion (S10, Shanghai Tuohe Electromechanical Technology Co., Ltd. Shanghai, China) equipped with a 3.2 mm diameter rotor. The mixture in a 10 mL centrifuge tube was subjected to high-speed shearing at 12,000 rpm and 4 °C for 2 min to prepare Pickering emulsions, which were screened based on various indicators, including particle size, zeta potential, and morphology.

### 2.10. Determination of Emulsion Droplet Size Distribution and Zeta Potential

The D(4,3) particle size of the emulsion was measured using a Malvern MS2000 laser particle size analyzer (Malvern Instruments Ltd., Malvern, UK), following the previously reported method [30]. Measurement conditions were as follows: water served as the dispersion medium, a temperature of 25 ± 1 °C, and a refractive index of 1.414. Samples were diluted with deionized water to 1 mg/mL, equilibrated for 120 s, and measured three times. The zeta potential of emulsion droplets was measured using a Nano-ZS nanoparticle analyzer (Malvern Instruments Ltd., Malvern, UK) following dilution with deionized water. Measurements were performed in the instrument’s standard capillary flow cell at 25 ± 1 °C, with an equilibration time of 60 s before three automatic parallel measurements.

### 2.11. Morphological Characterization and Microstructure Analysis of Emulsions

Emulsion appearance: Freshly prepared Pickering emulsion (6 mL) was loaded into a 10 mL centrifuge tube or transparent sample vial for photographic documentation.

Fluorescence microstructure: The emulsion microstructure was observed using an inverted fluorescence microscope. The prepared emulsion was diluted 2-fold with PBS, and the aqueous phase was stained with sodium fluorescein at a concentration of 1 mg/mL. A mixture of 50 μL of sodium fluorescein and 300 μL of the emulsion was incubated in the dark for 20 min. Subsequently, 10 μL of the sample was placed on a glass slide, examined under the microscope, and photographed.

### 2.12. Confocal Laser Scanning Microscopy (CLSM) Analysis

The prepared emulsion was diluted twice with PBS. Proteins were stained with 0.1% (*w*/*v*) Fast Green FCF (dissolved in distilled water), and the oil phase was stained with 0.1% (*w*/*v*) Nile red (dissolved in DMSO). Then, 50 μL of each dye was mixed with 300 μL of emulsion and incubated for 30 min. CLSM was used to analyze the microstructure of the Pickering emulsion at excitation wavelengths of 633 nm and 488 nm under 20× magnification.

### 2.13. Emulsion Stability Test

#### 2.13.1. Storage Stability and Creaming Index (CI) Analysis

Freshly prepared emulsion samples were stored at 4 °C or 25 °C. Photographic records of appearance changes and phase separation were taken monthly from 0 to 4 months. If phase separation occurs, the CI can be calculated as follows:$$\mathrm{CI}\left(\%\right)\;=\;(H_{1}/H_{2})\;\times \;100$$
where H_1_ is the clear liquid height (mm), and H_2_ represents the total emulsion height (mm).

#### 2.13.2. Centrifugation Stability

The centrifugation stability of Pickering emulsions was characterized by their water/oil-holding capacity. A 6 mL emulsion sample was centrifuged at 6000× *g* for 30 min at 4 °C. After centrifugation, the released water at the bottom and the extracted oil at the top of the tube were collected and measured. Water/oil-holding capacity was calculated as follows:$$\mathrm{Water}/\mathrm{Oil}\text{-}\mathrm{Holding}\;\mathrm{Capacity}\;=\;\frac{W_{1}-W_{2}-W_{3}}{W_{1}}\;\times \;100$$
where *W*_1_ denotes the initial sample mass, *W*_2_ represents the mass of released water post centrifugation, and *W*_3_ represents the mass of extracted oil post centrifugation.

#### 2.13.3. Evaluation of Thermal Stability

Based on the screening process above, the stability of Pickering emulsions prepared with the optimal formulation was investigated. The emulsion samples were heated in water baths at different temperatures (4, 25, 55, 75, and 95 °C) for 20 min. After treatment, the samples were cooled to room temperature. Post-treatment stability was evaluated by examining appearance, fluorescence microstructure, average particle size, and zeta potential.

#### 2.13.4. Evaluation of Salt Ion Concentration Stability

A salt stability test was conducted on Pickering emulsions prepared with the optimal formulation. NaCl stock solutions were pre-formulated at specific concentrations and quantitatively introduced into the emulsion system to achieve final NaCl concentrations of 0, 150, 300, 450, and 600 mmol/L. The emulsions were allowed to equilibrate for 1 h. Emulsions with varying ion concentrations were evaluated by examining morphology, fluorescence microstructure, average particle size, and zeta potential.

#### 2.13.5. Evaluation of pH Stability

Pickering emulsions with different initial pH values were prepared separately. First, the TBP–SA complex aqueous solution (dissolved in 0.01 M PBS, pH 7.0) was prepared. The pH of the aqueous phase was then precisely adjusted to 3, 5, 7, and 9 using 1 mol/L NaOH or 1 mol/L HCl. The pH-adjusted solutions were allowed to equilibrate for 1 h and then emulsified with MCT oil (φ = 60%) (using the method described in Section 2.9). Emulsions at different pH levels were characterized by examining appearance, fluorescence microstructure, average particle size, and zeta potential.

### 2.14. Data Analysis and Statistics

All experimental data in this study are expressed as mean ± standard deviation (SD) from three independent preparations of each sample. Graphical representations were generated using Origin 2025 and GraphPad Prism 9.0, while statistical analyses were performed via SPSS 27.0 using analysis of variance (ANOVA). If significant differences were detected (*p* < 0.05), post hoc multiple comparisons were conducted using Tukey’s Honest Significant Difference (HSD) test.

## 3. Results and Discussion

### 3.1. Optimization of TBP–SA Complex Preparation Conditions

#### 3.1.1. Effect of pH on Particle Size and Zeta Potential of TBP–SA Complex

At a fixed SA concentration of 0.2%, Figure 1 shows the effects of pH on the particle size and zeta potential of the TBP–SA complex. Under acidic (pH 3) and alkaline (pH 11) conditions—far from the isoelectric point of TBP (pH 4.0–5.0)—both particle size and zeta potential of TBP–SA and TBP significantly decreased and increased in absolute values, respectively. This phenomenon was attributed to the few surface charges of TBP near its isoelectric point, which weakened electrostatic repulsion between proteins and promoted aggregation [14,31]. At pH levels below the isoelectric point, the amino groups in TBP were protonated, while at pH levels above it, the carboxyl group ionization intensifies [32]. Compared with TBP alone, the TBP–SA complex consistently exhibited smaller particle sizes, reaching a minimum at a pH of 9.0. Its zeta potential absolute values exceeded those of pure TBP and peaked at pH 9.0. The inverse correlation between zeta potential magnitude and particle size trends was attributed to SA, which is an anionic polysaccharide that suppressed protein aggregation by modifying electrostatic repulsion on the TBP surface [13] and facilitated the formation of more compact complexes [33]. In summary, the addition of SA significantly enhanced the solubility and dispersion stability of TBP across a broad pH range (particularly near its isoelectric point), effectively inhibiting aggregation and precipitation. This improvement mainly stems from (1) enhanced electrostatic repulsion caused by negatively charged SA groups and (2) steric hindrance provided by SA molecular chains [22]. Given the minimal particle size and the maximal absolute value of zeta potential at a pH of 9.0, this pH was identified as optimal for TBP–SA complex preparation.

#### 3.1.2. Effect of SA Concentration on Particle Size and Zeta Potential of TBP–SA Complex

At pH 9.0, the SA concentration significantly influenced the properties of TBP–SA complexes (Figure 2). The particle size of TBP–SA complexes initially decreased and then increased, reaching a minimum at a 0.2% (*w*/*v*) SA. At lower SA concentrations (<0.2% (*w*/*v*)), larger particle sizes were observed, possibly due to insufficient SA coverage on TBP surfaces failing to fully shield hydrophobic interactions between particles, leading to aggregation. When SA concentration exceeded 0.2% (*w*/*v*), particle size increased, possibly due to insufficient TBP binding sites for effective interfacial interaction [34]. Notably, all TBP–SA complexes exhibited larger particle sizes than pure TBP across the tested SA concentrations. This phenomenon may be attributed to weak hydrophobic interactions and hydrogen bonding between SA and TBP, forming an SA coating on TBP surfaces [35,36,37]. These findings are consistent with existing studies. For example, increasing corn fiber gum (CFG) content from 0% to 12% enlarged pea protein/CFG complex particle size from 505 nm to 1354 nm [38].

The zeta potential absolute values of TBP–SA complexes initially increased and then stabilized as SA concentration increased, peaking at 0.2% (*w*/*v*) SA. This trend is attributed to SA—an anionic polysaccharide—enhancing surface negative charges on TBP. At 0.2% (*w*/*v*) SA, a uniform and dense coating maximized the exposure of anionic sites, thereby optimizing electrostatic repulsion [33]. Optimization Conclusion: At 0.2% (*w*/*v*) SA, minimal particle size and maximal zeta potential absolute values were simultaneously achieved, indicating optimal dispersion and electrostatic stability. Therefore, 0.2% (*w*/*v*) SA was used as the optimal concentration. Therefore, the optimal preparation conditions for TBP–SA complexes were as follows: temperature: 25 °C, TBP concentration: 1.0% (*w*/*v*), SA concentration: 0.2% (*w*/*v*), pH: 9.0. All subsequent characterizations of the complexes (e.g., surface hydrophobicity, SEM, DSC, FTIR) were performed on samples prepared and freeze-dried under these optimal conditions.

### 3.2. Analysis of the Emulsification Properties of TBP–SA Complex

The emulsification properties of the TBP–SA complex prepared under optimized conditions (pH 9.0, TBP 1.0% (*w*/*v*), SA 0.2% (*w*/*v*)) are shown in Figure 3. The results showed that the addition of SA significantly improved the emulsification performance of TBP. The emulsification activity index (EAI) increased from 35.98 ± 0.99 m^2^/g to 53.76 ± 1.76 m^2^/g, and the emulsification stability index (ESI) rose from 51.99 ± 1.47% to 78.78 ± 1.79%. The inherently poor emulsification properties of TBP were attributed to its molecular structural characteristics. TBP, a typical plant protein, is rich in hydrophilic amino acids with a compact conformation, resulting in low surface hydrophobicity and limited molecular flexibility. These properties hinder rapid adsorption at oil–water interfaces, molecular unfolding, and the formation of stable interfacial films [21,39]. Strong intermolecular interactions facilitated aggregation in solution, further reducing interfacial activity [36,40]. The significantly enhanced emulsification after forming TBP–SA complexes with 0.2% (*w*/*v*) SA involved three mechanistic pathways:

(1) Hydrophobic interactions and hydrogen bonding between SA and TBP induced partial unfolding of TBP, exposing internal hydrophobic groups and enhancing affinity for oil phases. This exposure facilitated interfacial adsorption [21,41]. (2) Steric hindrance: the long-chain structure of SA provided spatial hindrance, suppressing excessive TBP aggregation and improving migration efficiency to interfaces [21]. (3) Electrostatic repulsion: abundant carboxylate groups (-COO−) on SA chains increased negative charge density on complex surfaces, strengthening electrostatic repulsion between droplets and enhancing emulsion stability [36,37]. This outcome demonstrates the effectiveness of the polysaccharide–protein complexation strategy in enhancing plant protein emulsification, providing critical foundational data for developing novel food emulsification systems based on TBP–SA complexes.

### 3.3. Analysis of Surface Hydrophobicity

The surface hydrophobicity test results of TBP and TBP–SA complex are shown in Figure 4. Compared with pure TBP, the surface hydrophobicity of TBP–SA complex was significantly improved. This result directly confirms that the internal hydrophobic groups of TBP molecules are exposed during the formation of the complex. We speculate that the introduction of SA is primarily driven by hydrophobic interactions, inducing partial unfolding of TBP molecules and exposing their internal hydrophobic groups [32,42,43]. Meanwhile, the formation of a hydrogen bond network (FTIR results) stabilizes this unfolded conformation. The synergy of both ultimately leads to the formation of a complex with enhanced hydrophobic properties.

### 3.4. DSC Analysis of TBP–SA Complex

Figure 5 shows DSC curves of TBP, SA, and the TBP–SA complex. Pure TBP exhibited an endothermic peak at 80.33 °C, while pure SA exhibited its peak at 100.04 °C. The thermal denaturation temperature of the TBP–SA complex increased to 83.00 °C compared with pure TBP, showing a trend of enhanced thermal stability. This change may indeed be due to the following factors: First, an extensive hydrogen bond network formed between the hydroxyl and carboxyl groups on the SA molecular chains and polar groups such as amide groups on the TBP molecular surface. These strong intermolecular interactions create an effective “cross-linking effect” between TBP and SA, thereby stabilizing the native protein structure [32,44]. Second, Steric confinement: the long-chain SA molecules physically restrict conformational unfolding of TBP through spatial hindrance [36,45]. Therefore, the thermal denaturation temperature of the TBP–SA complex shows an increasing trend, indicating superior applicability in thermally processed food emulsions.

### 3.5. FTIR Analysis of TBP–SA Complex

FTIR spectra of TBP, SA, and the TBP–SA complex are shown in Figure 6. Characteristic peaks of SA were observed at a broad peak at 3417 cm^−1^, corresponding to O-H stretching vibration, while peaks at 1613 cm^−1^ and 1419 cm^−1^ corresponded to asymmetric and symmetric stretching vibrations of carboxylate groups (-COO−), respectively [46]. TBP featured an amide I band (1652 cm^−1^, C=O stretching vibration) and an amide II band (1540 cm^−1^, N-H bending vibration) [47].

Significant band shifts and peak shape changes were observed in the TBP–SA complex. The asymmetric stretching vibration peak of -COO− of SA blue -shifted from 1613 to 1639 cm^−1^, while its symmetric vibration peak red-shifted from 1419 to 1406 cm^−1^ [48]. The O-H stretching vibration peak (3417 cm^−1^) of SA blue-shifted and broadened to 3434 cm^−1^, while the N-H peak (3300 cm^−1^) of TBP was significantly weakened. These characteristic peak shifts indicate significant changes in their chemical environment, most likely due to the formation of hydrogen bond networks between hydroxyl groups of SA and amide groups (-CONH-) of TBP [48,49], which affected their vibrational energy levels. The amide I band of TBP red-shifted from 1652 to 1639 cm^−1^, indicating altered hydrogen bonding and possibly accompanying changes in protein secondary structure [50,51].

These changes (particularly carboxyl red-shift, hydroxyl blue-shift, and amide band displacement) confirm that the TBP–SA complex is formed through hydrogen bonding, which induces conformational changes. These molecular-level interactions corroborate the enhanced thermal stability revealed by DSC and the dense network structure revealed by SEM, collectively elucidating the stabilization mechanism of the complex.

### 3.6. SEM Results of TBP–SA Complex

SEM results of TBP and the TBP–SA complex are shown in Figure 7. SA exhibited aggregated flaky and reticulated porous structures [40]. TBP exhibited an amorphous flaky structure, consistent with that reported by Zhang and Zhuo [52,53]. The TBP–SA complex morphology reveals significant microstructural changes. SA was uniformly dispersed within the TBP matrix, forming a composite with a rough, dense network-like structure and significantly enhanced surface compactness. The dense network structure observed in the dry state suggests that the TBP and SA molecular chains likely achieve tight entanglement and assembly through hydrogen bonding and induced conformational changes [41].

### 3.7. Construction of Pickering Emulsions

We investigated the effects of different TBP–SA complex concentrations and oil phase percentages on the preparation of Pickering emulsions. Experimental results showed that variation in complex concentrations and oil phase ratios significantly affected the stability of Pickering emulsions, which was attributed to the synergistic effects of interfacial tension and particle adsorption capacity [54]. The appearance of Pickering emulsions is shown in Figure 8a. At low TBP–SA complex concentrations (e.g., 0.5% (*w*/*v*) and 1% (*w*/*v*)), emulsions prepared with oil phases ranging from 20% (φ) to 80% (φ) were unstable and exhibited significant phase separation. This instability is mainly attributed to the limited ability of low-concentration complexes to reduce interfacial tension and insufficient particle adsorption (coverage below the critical value), preventing the formation of effective mechanical barriers [55]. When the TBP–SA complex concentration was increased (e.g., 3% (*w*/*v*) and 5% (*w*/*v*)), the phase separation in Pickering emulsions significantly improved. At a fixed TBP–SA complex concentration of 5% (*w*/*v*), increasing the oil phase ratio led to the gradual formation of stable Pickering emulsions. The emulsion with 60% (φ) oil phase appeared uniformly milky and exhibited significantly enhanced stability, indicating that high-concentration TBP–SA composite particles effectively reduced oil–water interfacial tension [56]. This phenomenon might be attributed to the irreversible adsorption of TBP–SA composite particles at the oil–water interface, with hydrophobic regions embedded in the oil phase and hydrophilic regions extending into the aqueous phase, forming mechanically robust interfacial films [49]. Additionally, the high interfacial packing density of TBP–SA particles provided steric hindrance against coalescence, and this inference will be further verified in subsequent environmental stability tests of the emulsion [57]. Particle size and zeta potential data further support this observation. As shown in Figure 8b,c, when complex concentration increased from 0.5% (*w*/*v*) to 5% (*w*/*v*), particle size [D(4,3)] significantly decreased from 79.11 ± 0.71 μm to 29.49 ± 0.83 μm, mainly due to reduced interfacial tension promoting droplet dispersion [58]. Zeta potential significantly decreased from −29.39 ± 1.18 mV to −42.39 ± 0.71 mV, attributed to enhanced electrostatic repulsion between particles, resulting in tighter adsorption layers [59]. When oil phase concentration increased from 20% (φ) to 80% (φ), the particle size [D(4,3)] of Pickering emulsions initially decreased and then increased, while zeta potential initially decreased and subsequently increased.

Figure 8d shows laser confocal microscopy (CLSM) images. The stained oil phase exhibited red fluorescence, while dyed green protein particles densely adsorbed on the surface of the red-fluorescent oil droplet. The interfacial particle coverage area fraction was approximately 28.09% ± 6.04%. Further observation revealed two main distribution states of TBP–SA: one involving adsorption onto droplet surfaces, and the other forming connections with adjacent droplets through structural networks. This network-like interfacial structure indicates that the emulsion was stabilized by TBP–SA particles. The stability stems not only from particle adsorption but also from the powerful spatial confinement effect provided by this unique network [60]. When the oil phase ratio and complex concentration were 60% (φ) and 5% (*w*/*v*), respectively, the prepared emulsion displayed bright, vivid coloration with fine and uniformly distributed particles, indicating that TBP–SA particles adequately covered the oil phase and effectively enhanced emulsion stability. Based on a comprehensive analysis of appearance morphology, particle size [D(4,3)], and zeta potential, the optimal ratio for preparing Pickering emulsions was determined to be 5% complex concentration and 60% oil phase ratio.

### 3.8. Physical Stability of Pickering Emulsions

#### 3.8.1. Storage Stability

The storage stability results are shown in Figure 9. No phase separation was observed in the Pickering emulsions during the 4-month storage period, indicating excellent storage stability. This finding shows that higher concentrations of TBP–SA complexes contribute to emulsion stabilization. This phenomenon may be attributed to the gel network structure of the complexes and the increased solution viscosity brought by SA, where the three-dimensional network system effectively restricts the free movement of oil droplets, inhibit droplet collision, prevent flocculation, thereby enhancing the physical stability of the emulsion [61,62].

#### 3.8.2. Centrifugal Stability

The centrifugal stability of emulsions is a crucial indicator for evaluating the physical stability of emulsions. By simulating accelerated separation processes, it predicts emulsion stability during storage and use, playing a significant role in assessing functional performance in food applications. The centrifugal stability results are shown in Figure 10. No significant changes or phase separation was observed in the appearance of Pickering emulsions, indicating good centrifugal stability. This result was confirmed by the water-holding and oil-holding capacity of 96.33 ± 1.48%, indicating that only a small amount of oil and water was separated from the emulsion system. This phenomenon indicates that TBP–SA complexes exhibit excellent interfacial adsorption capacity, mainly attributed to the contribution of dense interfacial films and SA-induced weak gel networks (aqueous phase thickening or interfacial networks) to restricting oil droplet movement, thereby endowing Pickering emulsions with excellent stability [35,61].

### 3.9. Environmental Response Stability of Pickering Emulsions

#### 3.9.1. Thermal Stability

The thermal stability of Pickering emulsions was characterized multidimensionally, and the results are shown in Figure 11. When the Pickering emulsions were treated at temperatures ranging from 4 °C to 75 °C, the emulsion system exhibited distinct gelation characteristics, further confirmed by inversion tests (Figure 11a). Inverted fluorescence microscopy (Figure 11b) revealed uniformly distributed fine particles in Pickering emulsions with slight aggregation, indicating stable performance across a broad temperature range. As the temperature increased to 95 °C, the particle size distribution [D(4,3)] significantly increased from 29.49 ± 0.83 μm to 35.39 ± 0.56 μm (Figure 11c). Additionally, the absolute value of the zeta potential decreased to some extent (−42.39 ± 0.71 mV → −37.17 ± 0.33 mV) (Figure 11d). These parameter variations indicated that high-temperature treatment promoted droplet aggregation, reduced system stability, and weakened hydrogen bonds, leading to interfacial film relaxation. Nevertheless, the emulsion maintained good macroscopic homogeneity without visible phase separation, remaining within acceptable limits. A study on soluble starch/whey protein isolate-stabilized Pickering emulsions similarly reported reduced thermal stability after treatment, as evidenced by increased fluidity and significantly enlarged droplet size [63].

#### 3.9.2. Salt Ion Concentration Stability

When the ion concentration increased from 0 mM to 600 mM, the Pickering emulsions showed no obvious stratification or creaming, presenting a non-flowing, gel-like morphology (Figure 12a). Inverted fluorescence microscopy (Figure 12b) revealed a gradual decrease in uniformity and an increase in particle size in Pickering emulsions; the particle size [D(4,3)] significantly increased from 29.49 ± 0.83 μm to 36.13 ± 0.84 μm (Figure 12c). The absolute value of the zeta potential of the Pickering emulsion decreased(−42.39 ± 0.17 mV → −37.13 ± 0.73 mV) (Figure 12d). The possible mechanism lies in the high salt concentration compressing the electrical double layer, reducing electrostatic repulsion, enhancing van der Waals attraction between particles, and promoting aggregation to form a three-dimensional network structure, thus exhibiting gel-like behavior and a thickening effect macroscopically. The salt-induced aggregation and structuring phenomenon maintained macroscopic stability (inhibiting phase separation) while promoting particle growth through interfacial particle rearrangement, representing a typical dual electrical layer regulation-steric hindrance synergistic stabilization effect [54].

#### 3.9.3. pH Stability

At pH values of 3.0 and 5.0, the emulsions were liquid and flowed freely after inversion, while at pH values of 7.0 and 9.0, they formed self-supporting, non-flowing, gel-like structures (Figure 13a). Inverted fluorescence microscopy (Figure 13b) revealed gradually reduced uniformity and increased particle size in Pickering emulsions at pH values of 3.0 and 5.0, while at pH values of 7.0 and 9.0, they exhibited better uniformity and smaller particles. Near the isoelectric point, the pH 5.0 sample exhibited the lowest zeta potential (−26.12 mV) and largest particle size [D(4,3)] (79.81 μm) (Figure 13c,d), indicating that reduced electrostatic repulsion led to particle aggregation. Away from the isoelectric point, a pH of 3.0 formed loose interfaces due to partial protonation [64]. In contrast, pH values of 7.0 and 9.0 significantly suppressed aggregation through enhanced negative charges. At a pH of 7.0, the emulsions exhibited the smallest particle size, possibly due to synergistic stabilization caused by electrostatic repulsion and SA gel networks. Although the emulsions maintained high zeta potential at a pH of 9.0, their particle size slightly increased at a pH of 7.0, possibly due to conformational changes in the SA chain under strong alkalinity, but overall conforming to the pattern: lowest potential → largest size → isoelectric destabilization.

## 4. Conclusions

In this study, wide pH-stable Pickering emulsions were successfully developed using TBP–SA composite particles, and the optimal preparation conditions for the composite were identified. The composite particles formed a dense network structure through hydrogen bonding and hydrophobic interactions, which significantly improved emulsification performance and thermal stability, with the thermal denaturation temperature also showing an increasing trend. The emulsion exhibited good macroscopic physical stability (i.e., no phase separation), maintaining integrity and homogeneity across a wider pH range (3.0–9.0), high salt conditions (≤600 mM), and short-term high temperature (≤95 °C). At the microscopic and rheological level, its behavior varied with the environment: under optimal pH (7.0–9.0) and high salt conditions, the emulsion exhibited a stable gel state (non-flowing), whereas near the protein isoelectric point (pH 3.0–5.0) or at 95 °C high temperature, although droplet aggregation increased fluidity, the robust interfacial film effectively prevented demulsification. The stabilization mechanisms included (1) the high zeta potential of TBP–SA particles, which provided strong electrostatic repulsion; (2) the dense interfacial network, which enhanced mechanical strength; and (3) SA-induced gelation, which restricted droplet mobility. This system offered a novel carrier for delivering active ingredients.

The strength of this study lies in the successful construction and systematic characterization of a novel Pickering emulsion stabilization system based on natural plant proteins and polysaccharides, which exhibits good environmental stability. However, the study also has some limitations, such as the lack of in-depth investigation into the direct mechanical properties of the interfacial adsorption film (e.g., interfacial rheology). Future research will focus on using techniques like cryo-electron microscopy to directly observe the interfacial structure and exploring the actual loading and controlled release performance of this delivery system for specific active ingredients (e.g., vitamins, polyphenols). In terms of commercial applications, this TBP–SA-based Pickering emulsion has potential application prospects in developing highly stable functional beverages, sauces, meal replacement foods, and cosmetic emulsions.

## Figures and Tables

**Figure 1 foods-14-03429-f001:**
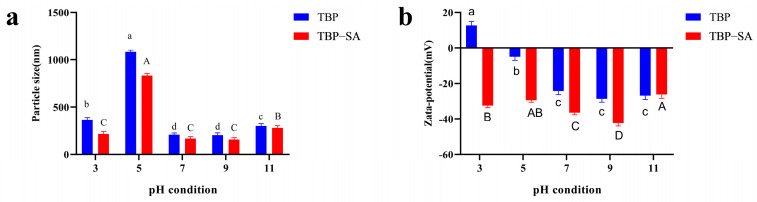
Particle size and zeta potential of TBP–SA composite particles under different pH conditions ((**a**) Particle size distribution; (**b**) Zeta potential). Note: Different letters indicate significant differences (*p* < 0.05) between sample mean values; lowercase letters (a–d) compare TBP and uppercase letters (A–D) compare TBP–SA particles across different pH conditions. the same applies below.

**Figure 2 foods-14-03429-f002:**
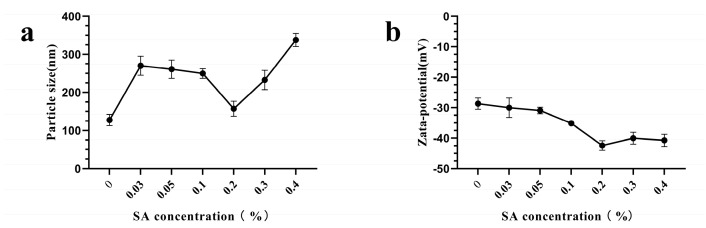
Effects of SA concentration on particle size and zeta potential of TBP–SA composite particles ((**a**) Particle size distribution; (**b**) Zeta potential).

**Figure 3 foods-14-03429-f003:**
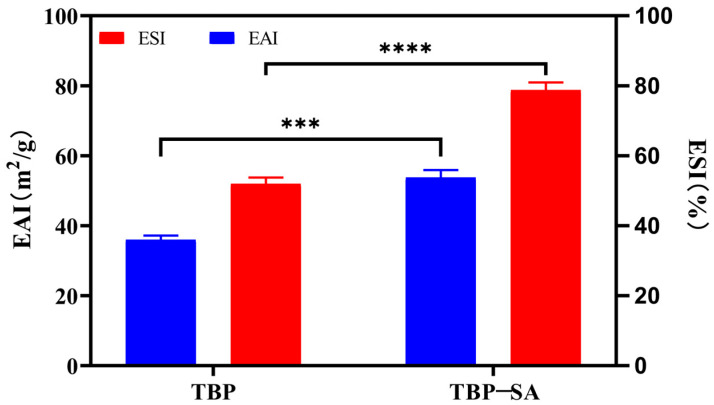
Emulsifying activity index (EAI) and emulsifying stability index (ESI) of TBP and TBP–SA composite particles. Note: *** represents *p* < 0.001, **** represents *p* < 0.0001.

**Figure 4 foods-14-03429-f004:**
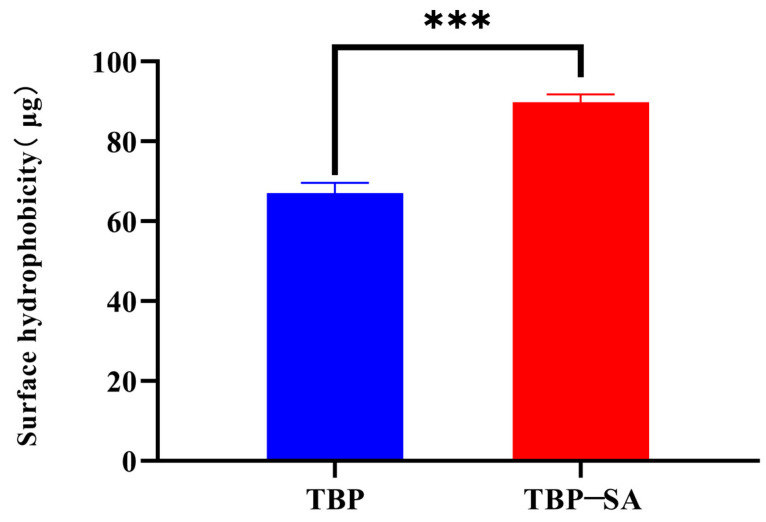
Surface hydrophobicity analysis of TBP and TBP–SA complex. Note: *** represents *p* < 0.001.

**Figure 5 foods-14-03429-f005:**
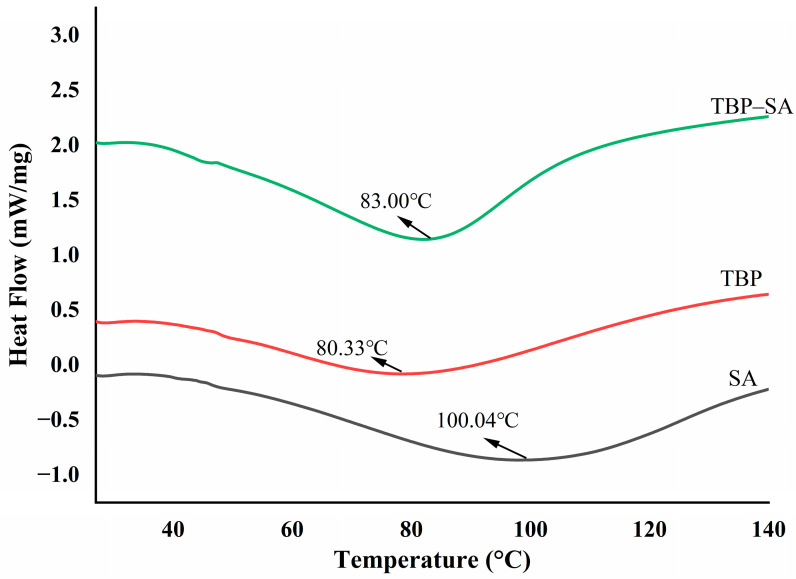
Thermal denaturation properties of TBP and TBP–SA composite particles (DSC).

**Figure 6 foods-14-03429-f006:**
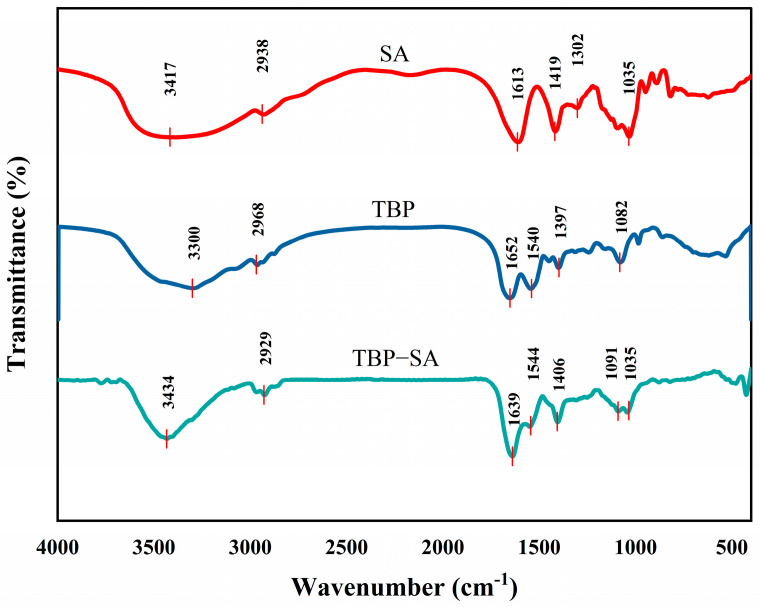
Chemical bond characteristics of TBP and TBP–SA composite particles (FTIR spectra).

**Figure 7 foods-14-03429-f007:**
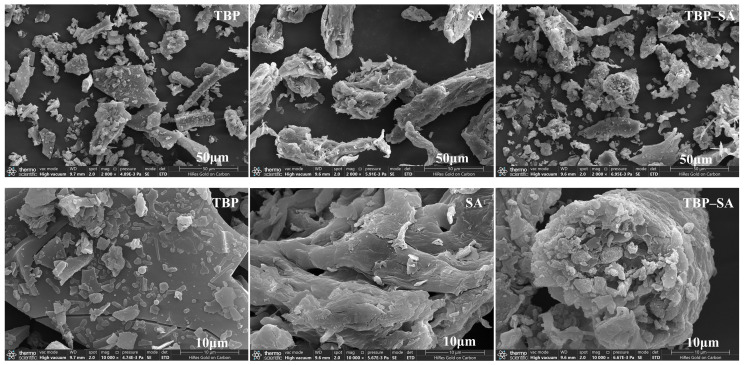
Microstructure of TBP and TBP–SA composite particles (SEM).

**Figure 8 foods-14-03429-f008:**
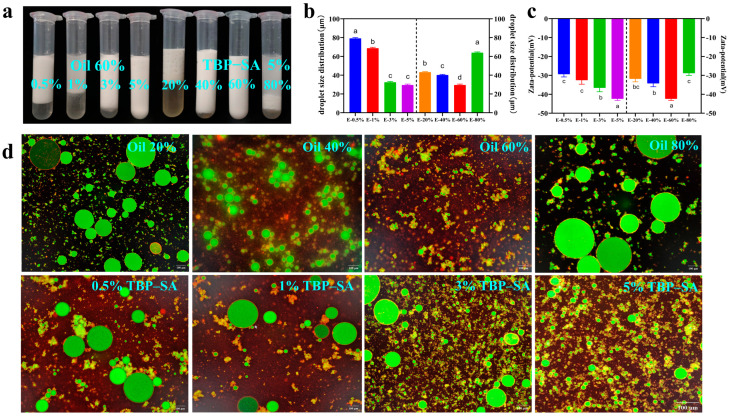
Characterization of Pickering emulsions stabilized by TBP–SA composite particles ((**a**) macroscopic appearance; (**b**) droplet size distribution; (**c**) zeta potential; (**d**) interfacial adsorption morphology by CLSM). The letters indicate significant differences (*p* < 0.05).

**Figure 9 foods-14-03429-f009:**
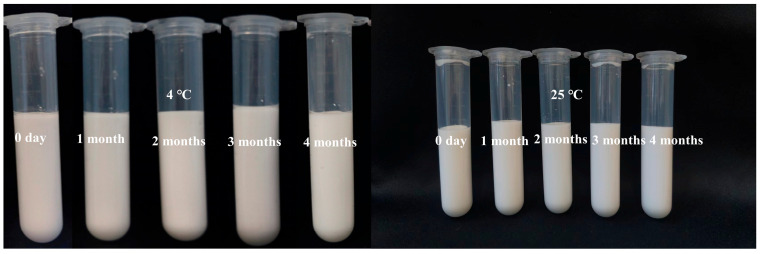
Storage stability of Pickering emulsions over 4 months.

**Figure 10 foods-14-03429-f010:**
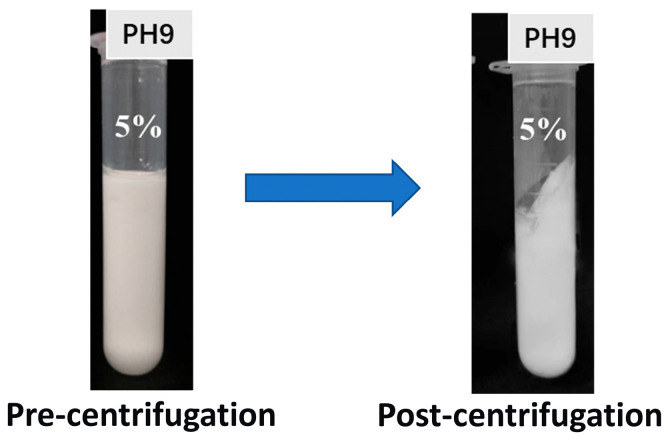
Centrifugal stability of Pickering emulsions.

**Figure 11 foods-14-03429-f011:**
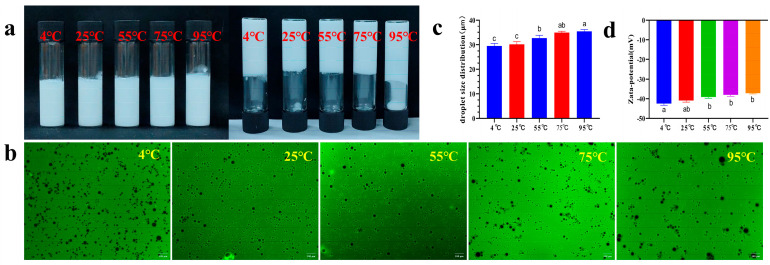
Thermal stability of Pickering emulsions ((**a**) Macroscopic appearance, The upright test tube shows the macroscopic homogeneity of the emulsion, and the inverted test tube is used to evaluate its self-supporting ability or fluidity to visually characterize its gel strength/physical stability; (**b**) fluorescence microscopy; (**c**) droplet size distribution; (**d**) zeta potential). The letters indicate significant differences (*p* < 0.05).

**Figure 12 foods-14-03429-f012:**
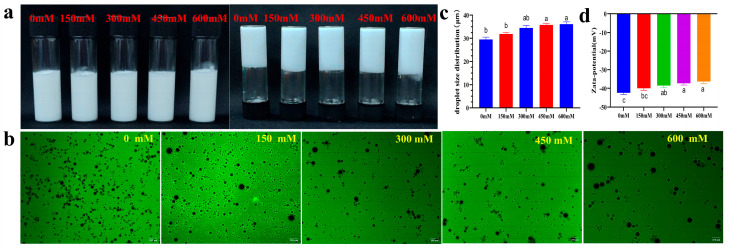
Salt ion stability of Pickering emulsions ((**a**) macroscopic appearance, the upright test tube shows the macroscopic homogeneity of the emulsion, and the inverted test tube is used to evaluate its self-supporting ability or fluidity to visually characterize its gel strength/physical stability; (**b**) fluorescence microscopy; (**c**) droplet size distribution; (**d**) zeta potential). The letters indicate significant differences (*p* < 0.05).

**Figure 13 foods-14-03429-f013:**
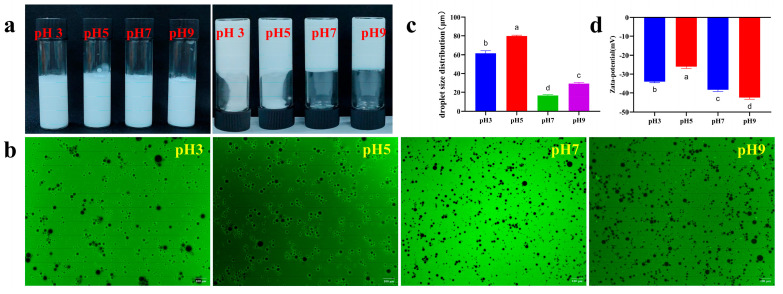
pH stability of Pickering emulsions ((**a**) macroscopic appearance, the upright test tube shows the macroscopic homogeneity of the emulsion, and the inverted test tube is used to evaluate its self-supporting ability or fluidity to visually characterize its gel strength/physical stability; (**b**) fluorescence microscopy; (**c**) droplet size distribution; (**d**) zeta potential). The letters indicate significant differences (*p* < 0.05).

## Data Availability

The original contributions presented in the study are included in the article, further inquiries can be directed to the corresponding authors.

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
