# Peer review of "Fabrication and Mechanism of Pickering Emulsions Stability over a Broad pH Range Using Tartary Buckwheat Protein–Sodium Alginate Composite Particles"

_foods, 2025, doi:10.3390/foods14193429_

Round 1

Reviewer 1 Report

Comments and Suggestions for Authors

The manuscript is about TBP–SA complexes (pH 9, TBP 1 wt%, SA 0.2 wt%), report improved EAI/ESI, modestly higher DSC denaturation temperature, and construct O/W Pickering emulsions (5 wt% complex, 60 % oil) that resist creaming over 4 months at 4 °C and tolerate brief heat/salt/pH stresses, attributing stability to high |ζ|, interfacial particle networks, and SA‑driven gelation 

I recommend some revisions. 

Introduction

Line 44, please consider adding a focused gap analysis including the following topics

(i) what is known on protein–alginate complexes in Pickering systems;
(ii) specific deficiencies for TBP (e.g., low surface hydrophobicity, aggregation near pI);
(iii) what remains unknown (interfacial mechanics across pH/ionic strength; gelation role of SA); and
(iv) your testable hypothesis for TBP–SA at pH 9.

Line 60, it would be beneficial to nuance “no study reported” by specifying scope (TBP–SA particles for Pickering) and the distinct mechanistic question you test.

Materials

Line 73, please, resolve the contradiction “Laboratory‑prepared TBP … purchased from Shanghai Aladdin.” State clearly whether TBP was self‑isolated (give method) or purchased; if prepared, provide the protein isolation protocol and final purity assay used (Kjeldahl/BCA).

Methods – Complex preparation & characterization

Line 83, I recommend, reporting sonication details (probe diameter, amplitude %, pulse on/off, vessel volume, temperature control) because they strongly affect complexation and dispersion.

Line 86, please, clarify whether particle size/ζ‑potential were measured on re‑dispersed powders post‑freeze‑drying; if so, describe rehydration conditions (time, ionic strength, pH, sonication). Measuring “after freeze‑drying” without redispersion detail is not reproducible.

Line 126, please, add SEM conditions (accelerating voltage, working distance, detector, magnification range) and state whether images show freeze‑drying artifacts; SEM of powders does not prove interfacial film structure—tone down mechanistic claims or add cryo‑SEM/AFM of interfaces.

Methods – Emulsion preparation & measurements

Line 133, please, specify the homogenizer model/rotor‑stator head, cup geometry, gap, and energy density (W·s/mL). Report temperature control during shearing and rest time before measurements.

Line 135, please consider stating the exact ionic strength: you dissolve powders in PBS 0.01 mol/L (pH 7) then adjust to pH 9; this changes ionic strength. Salt strongly affects ζ and stability; report final conductivity/ionic strength used in each assay.

Line 147, please, detail ζ‑potential of emulsion droplets (instrument, cell type, dilution medium, ionic strength). Measuring in deionized water can artificially increase |ζ|; use a controlled electrolyte matching the sample matrix.

Line 169, please, correct the creaming‑index equation typography: make it explicit as CI (%) = (H₁/H₂)×100 and define measurement precision for H₁/H₂.

Line 181, please, specify whether samples were cooled to a common temperature before size/ζ measurements after heating; otherwise, apparent changes may reflect temperature dependence of viscosity/RI rather than stability.

Line 188, it would be beneficial to state whether NaCl was added on top of PBS (report final ionic strength) and the equilibration time before measurements.

Line 195, please, report the final buffer capacity at each pH (3, 5, 7, 9) and equilibration time; large pH jumps can transiently alter interfacial properties.

Line 199, please, specify the post‑hoc multiple comparison test (e.g., Tukey HSD) used to generate the letter groupings, and confirm the number of biological vs. technical replicates.

Results – Optimization and contradictions

Line 215, please, reconcile this statement (“TBP–SA complex consistently exhibited smaller particle sizes… minimum at pH 9”) with Line 239–241 (“all TBP–SA complexes exhibited larger particle sizes than pure TBP across SA concentrations”). Clarify that Fig. 1 compares across pH at fixed SA while Fig. 2 varies SA at pH 9; indicate which condition is used for all later comparisons.

Line 252, I recommend, either include the temperature‑effect data (25–100 °C) you mention or remove the claim; also justify why 25 °C is deemed “optimal” if higher temperatures “slightly improved” properties.

Line 289–299, please consider tempering the DSC interpretation: a 2.7 °C increase without statistics is marginal; present replicate DSC traces or assign uncertainty.

Line 309–318, please, avoid asserting β‑sheet enhancement from the amide‑I shift without spectral deconvolution/curve fitting; otherwise describe as “indicative of altered hydrogen bonding and possible secondary‑structure changes.”

Line 331–334, please consider reframing SEM observations as dry‑state morphology only, not proof of “dense network at the interface.” If you wish to keep the “mechanically robust interfacial film” claim, add interfacial shear rheology or cryo‑microscopy.

Line 352–359, please, substantiate the statement “interfacial packing density provided steric hindrance” with interfacial tension kinetics and/or adsorbed mass (e.g., drop tensiometry, QCM‑D), or present it as a hypothesis.

Line 368–377, it would be beneficial to add scale bars and quantify interfacial coverage (area fraction) in CLSM images; currently the evidence is qualitative.

Results – Stability

Line 386–395, I recommend, reporting storage stability also at 25 °C (and perhaps 37 °C) to broaden practical relevance; 4 °C only limits applicability.

Line 414–421, please, describe changes precisely: ζ‑potential became less negative (−42.39 → −37.17 mV) and droplet size increased; avoid saying “increased ζ‑potential” without specifying magnitude/sign.

Line 429–432 & Fig. 10b, please consider providing all five temperature micrographs (4, 25, 55, 75, 95 °C) in panel b; one panel appears missing/blank—ensure consistency and include scale bars.

Line 433–446, please, state whether the “gel‑like appearance” under salt is due to bulk gelation (continuous‑phase SA network) by adding small‑amplitude oscillatory rheology (G′, G″) across NaCl; otherwise, tone down “gelation” to “apparent thickening.”

Line 451–454, please, fix the contradiction: “the emulsions were liquid and flowed freely after inversion,” then “did not flow when inverted.” Decide which is correct (likely “self‑supporting at pH 3–5; free‑flowing at pH 7–9”).

Line 457–467, please consider discussing charge state consistency: at pH 9 both TBP (net negative above pI ≈ 4–5) and SA (carboxylates) are negative; thus electrostatic attraction cannot dominate. Emphasize hydrogen bonding/hydrophobic patches and residual cationic residues (Lys/Arg) as plausible contributors, or support electrostatics with titration/ζ‑potential of the complex vs. components.

Conclusions

Line 472–483, I recommend, aligning wording with corrected evidence: separate long‑term storage (4 °C, 4 months) from short‑time stress tests (heat ≤ 95 °C for 20 min; NaCl ≤ 600 mM; pH 3–9). Avoid implying simultaneous long‑term high‑T storage.

Reviewer 2 Report

Comments and Suggestions for Authors

The manuscript is well written. Please find comments below to improve. The resolution of the figures can be improved.

Line 82: Are these solutions wt/wt? Also proper way to write is e.g., 2% wt/wt. Please check and rewrite in an appropriate way in the entire manuscript.

Line 84-85, what %acid and %alkali did you use to adjust pH? Reason for ultrasonication?

Line 86: Did you measure particle size and zeta potential of the solution, or did you dissolve the freeze-dried powder back to measure particle size and zeta potential?

Line 111 and 112: Expand EAI and ESI at first use.

Line 125: How did you process FTIR data after the scans?

Line 138: Give the name of the equipment used for high-speed shearing dispersion.

Line 157: Expand CLSM.

Line 268: Any reason for limited molecular flexibility with TBP? How do we know about limited molecular flexibility?

Line 297-299: Compared ot pure TBP peak at 80.33C, TBP-SA complex showed peak at 83.0 C, which is much less than SA alone. Why do you think increasing 3 C, is a significant improvement in thermal stability?

Line 327-337& Figure 6: It is hard to see any potential difference between the SEM images; if it does not give any conclusive data, we can consider removing SEM. Also, SEM gives detail about microstructure, it can not tell aobut electrostatic attraction or hydrogen bonding.

Figure 7: The Resolution of the whole image can be improved for better clarity. 7d. Confocal images need a scale bar.

Line 387-393: Can you explain in detail how the viscosity of the solution helps in fat droplet aggregation, what kind of destabilization it is preventing?

Figure 10a,11a &12a. Give more details on what the Figure shows with the straight and inverted tube.

Conclusion: Please mention the strengths, limitations, and scope of the study, as well as the commercial application of the research.

Reviewer 3 Report

Comments and Suggestions for Authors

The manuscript entitled "Fabrication and Mechanism of broad-pH-stable Pickering emulsions stabilized using Tartary Buckwheat Protein-Sodium Alginate composite particles" is an important topic of interest for developing a potential delivery system for bioactives. The authors address some important questions regarding the physical stability of different formulations. The manuscript needs minor revision. Some comments need to be clarified. Some points are:

Page 5, Lines 195–197:
The statement “pH of Pickering emulsions was adjusted to…” requires clarification. It is unclear whether the authors are referring to the prior adjustment of the aqueous phase to pH 3, 5, 7, and 9 using NaOH and HCl solutions before emulsification. This should be explicitly stated to avoid ambiguity. Additionally, the authors should specify whether a buffer system—specifically PBS—was used to dissolve TBP-SA and, if so, what its concentration was. This information is important to assess whether the buffer capacity was sufficient to maintain pH stability during emulsification and subsequent measurements.

Page 13, Lines 412–428:
The authors should consider addressing the potential oxidation of the oil phase at high temperatures. Oxidative degradation can influence not only the chemical integrity of the oil but also the physical stability of emulsions over time. A brief discussion on the oxidative stability of the oil under the experimental conditions, and whether any measures were taken to mitigate oxidation (e.g., use of antioxidants, inert atmosphere), would enhance the completeness of the study.

Round 2

Reviewer 1 Report

Comments and Suggestions for Authors

Thank you for replying to all queries. I believe the work can be accepted. 

Author Response

Thank you very much for taking the time to review this manuscript.

Reviewer 2 Report

Comments and Suggestions for Authors

The authors made the required revisions in the revised manuscript as per the suggestions. There is only one suggestion for improvement.

Revised Figures with confocal microscopy (Figures 8, 11, 12, 13)- I still cannot see the scale bar in the revised images; either it is not placed or the images have not been revised. Please address this. Also, when exporting an image, keep a higher resolution for optimal clarity.

Author Response

Thank you very much for taking the time to review this manuscript.
We want to clarify a misunderstanding regarding the image quality for the reviewer.The full revised manuscript we submitted on September 27 contains all figures in high resolution with clear scale bars. The point-by-point response file we provided for the reviewers had to use lower-resolution images to comply with email attachment size limits.
We apologize for this oversight.